# Development and Validation of a Food Literacy Assessment Tool for Community-Dwelling Elderly People

**DOI:** 10.3390/ijerph18094979

**Published:** 2021-05-07

**Authors:** Hyeona So, Dahyun Park, Mi-Kyung Choi, Young-Sun Kim, Min-Jeong Shin, Yoo-Kyoung Park

**Affiliations:** 1Department of Gerontology, Graduate School of East-West Medical Science, Kyung Hee University, Yongin 17104, Korea; thgus3700@naver.com (H.S.); ysunkim@khu.ac.kr (Y.-S.K.); 2Interdisciplinary Program in Precision Public Health, Graduate School, Korea University, Seoul 02841, Korea; ekgus7171@korea.ac.kr; 3Faculty of Food and Health Sciences, College of Natural Sciences, Keimyung University, Daegu 42601, Korea; mkchoi@kmu.ac.kr; 4Department of Biosystems and Biomedical Sciences, College of Health Science, Korea University, Seoul 02841, Korea; 5Department of Medical Nutrition, Graduate School of East-West Medical Science, Kyung Hee University, Yongin 17104, Korea

**Keywords:** food literacy, elderly, validity, questionnaire

## Abstract

Food literacy refers to the knowledge, skills, and attitudes required for individuals to choose foods that promote health. As the rate of diet-related diseases increases, food literacy is becoming more important. However, there are no tools available to evaluate food literacy among the Korean elderly. We derived 547 questions from a literature review and, after three rounds of Delphi surveys, selected 33 preliminary questions. We calculated the content validity ratio of the questions and applied a face validity procedure. We then selected 32 questions, assessed their validity, and distributed them as a questionnaire to 205 elderly people. We then conducted exploratory factor analysis (EFA) to determine the validity of the questionnaire and used an internal consistency index (Cronbach’s α coefficient) to determine reliability. Based on the factor analysis, 13 questions were selected, distributed among three factors, and evaluated using the Kaiser–Meyer–Olkin (KMO) and Bartlett sphericity tests. The factor analysis showed that KMO was 0.872, which is a highly acceptable score, and the Bartlett sphericity test was χ^2^ = 1,374.69 at *p* = 0.00. The food literacy questionnaire developed in this study will likely be helpful for improving the healthcare of elderly people.

## 1. Introduction

Food literacy is a recently introduced concept pertaining to knowledge and skills related to the use and production of food. Components of food literacy include access to and the management of food, and the selection, planning, preparation, cooking and ingestion of meals [1,2]. Food literacy helps to protect and promote health via the choice of appropriate foods [3]. Food literacy also encompasses the application of nutritional information [4]. Food literacy can increase awareness of salt intake and plays a role in the onset of diet-related diseases. When food literacy is low, the likelihood of various chronic diseases increases, resulting in significant costs to individuals and society [5]. Greater food literacy is associated with more self-control, less impulsive behavior, consumption of more nutritious foods, and better health and well-being. Food literacy can thus prevent chronic diseases and facilitate disease management [2,6,7,8,9,10].

The Self-Perceived Food Literacy Questionnaire (SFLQ) [2] comprises 12 questions and is applied in subjects aged 15–65 years. The multi-dimensional Food and Nutrition Literacy (FNLIT) [11] construct accounts for the social, cultural, and political aspects of food and nutritional literacy. However, it was developed in the context of elementary school students and has not yet been applied to elderly people.

According to the World Population Outlook and Statistics Korea, a study by the U.N., the proportion of people aged 65 years or older in Korea is expected to increase by up to 47% by 2067 [12]. The Korean Medical Panel estimated that more than 9 out of 10 elderly people suffer from one or more chronic disease [13]. Therefore, managing chronic diseases in an aging society is of critical importance. Understanding the level of food literacy among the elderly, as well as providing appropriate interventions according to their food literacy level, is essential to prevent and manage diseases. Few studies have focused on how to promote good nutritional habits in vulnerable groups, especially older adults. Developing a food literacy scale for the elderly population could help maintain the health of, and prevent and manage chronic diseases in, older adults.

## 2. Materials and Methods

A thorough review of research and content analysis to identify previously validated questionnaires were conducted with a full literature search from the year 1960 to 2020. We then constructed a set of preliminary questions followed by an evaluation of the content and face validity of the selected questions. Finally, we evaluated the construct validity and finalized the questionnaire (Figure 1). All procedures were approved by the Institutional Review Board of Korea University (KUIRB-2019-0306-01), and all study participants provided informed consent. Linear regression analyses were conducted to confirm the relationship of food literacy scores with food knowledge and Nutrition Quotient for Elderly (NQ-E) scores.

### 2.1. Extraction of Preliminary Questions

The PubMed, Koreanstudies Information Service System (KISS), Research Information Sharing Service (RISS), and Web of Science databases were utilized, using the key words “food literacy,” “health literacy,” and “nutrition literacy.” The collected papers were classified by research topic (definitions, measurement tools, and utilization) both domestically and abroad, and, then, the main methods and results of each study were organized into a consistent framework. From the extracted questionnaire, we further selected questions that are more appropriate for the elderly. Two of the researchers performed the literature search independently; 5 of the researchers together developed the preliminary questionnaires obtained from the search.

### 2.2. Evaluation of Validity and Reliability

#### 2.2.1. Content and Face Validity

To confirm the legitimacy of the content of the preliminary questionnaire, three rounds of Delphi surveys were conducted by 15 experts in food and nutrition. The experts were chosen based on their expertise area (professors in the area of clinical nutrition, epidemiology, traditional foods and nutrition, and nutrition education; a nutritionist at a local community service area; and a nutritionist working at the government) with at least 10 years of work experience. The content validity ratio (CVR) was assessed using the following four-point Likert scale: very inappropriate, inappropriate, appropriate, and very appropriate. The content was considered sufficient if the value was >0.99 for 5 experts, 0.62 for 10 experts, and 0.42 for 20 experts [14]. For the assessment of face validity, 10 adults aged > 65 years were recruited through convenience sampling and interviewed to determine whether the questions were straightforward to answer. The CVR was calculated as follows (Equation (1)):(1)$$\mathrm{CVR}\;=\frac{N_{e}-\left(\frac{N}{2}\right)\;}{\frac{N}{2}}$$

#### 2.2.2. Construct Validity

To examine the construct validity, a survey was conducted on 205 community-dwelling senior citizens (aged ≥ 65 years or older across Korea between June and November 2020). The inclusion criteria was current residence in Korea with Korean nationality, and those without visual disturbance or severe vision impairment. Illiterate individuals and those who have difficulty in understanding the context and communication were also excluded. Data on general age, sex, education, and monthly income were collected.

The NQ-E, a tool for measuring dietary behavior and food literacy developed and validated by the Korean Nutritional Society, was used to validate the questionnaire. Scores on food knowledge questionnaires created by the researchers were correlated with the food literacy data of the elderly participants. Exploratory Factor Analysis (EFA) was conducted for the valuation of questions, and an internal consistency index examined reliability. The EFA was based on main spindle factor analysis and the varimax rotation scheme and was further evaluated using the widely used KMO and Bartlett sphericity tests. A Kaiser-Meyer-Olkin (KMO) value > 0.60 was considered acceptable. In order to confirm that the correlation matrix was an identity matrix, a Bartlett sphericity test was used as a verification tool [15]. SPSS Statistics (version 25.0; IBM Corp., Armonk, NY, USA) was utilized for all statistical analyses.

## 3. Results

### 3.1. Extraction of Preliminary Questions

We selected a total of 112 papers, and a total of 547 questions were obtained through the literature search. These questions were classified into eight food information categories (production, processing, distribution, planning and management, selection, preparation and cooking, ingestion, and disposal) and three information comprehension categories (functional, interactive, and critical) [1,3]. The 547 questions were also classified into 21 domains (three food literacy domains × seven food system domains) [16].

### 3.2. Evaluation of Validity and Reliability

#### 3.2.1. Content and Face Validity

Three rounds of Delphi surveys were performed. Keywords were derived, and some of the questions were rephrased according to expert feedback from Delphi surveys, while those with a low validation index (<0.62) were either omitted or modified. The average content validity ratio (CVR) for the final 32 questions was adequate, at 0.88 [17,18] (Table 1).

#### 3.2.2. Construct Validity

After the selected questionnaire was produced, we then distributed the questionnaire to community-dwelling elderly from June to November 2020. Two hundred and five participants finished the questionnaire. The general characteristics of the participants are shown in Table 2. The number of female participants who answered the survey (mean age of 71.80 ± 6.81 years) was greater than that of male participants (59.5% vs. 40.5%).

The most frequently reported level of education was “≤middle school” (55.6%). Nearly 81% of participants reported a monthly income of “<KRW 4,000,000”. Over half (55.1%) of participants were living in a household of two people.

The collected data were then used for EFA procedure. Based on the EFA results, each question was assigned to a factor; questions that did not meet the criteria (i.e., extraction communalities or factor loading < 0.4) were omitted (Table 3). The data were adequate based on the statistical test results (KMO = 0.872; χ^2^ = 1374.69, degrees of freedom = 78, *p* = 0.00). Finally, of the 32 questions, 19 were removed; the remaining 13 were divided into three factors: preparation and cooking (Cronbach’s α = 0.894), distribution (Cronbach’s α = 0.825), and production and disposal (Cronbach’s α = 0.710).

The nutrition quotient (NQ) provides a comprehensive evaluation of the nutritional status and quality of an individual or group [19,20]. The NQ-E, developed by the Korean Nutrition Society, is a validated tool for quickly identifying the dietary status of elderly people. The NQ-E can be used to determine meal and diet quality based on the scores in four domains: eating behavior, balance, diversity, and moderation [21]. The eating behavior section includes six items: discomfort when chewing food; health awareness; depressive symptoms; habitual hand washing before meals; appropriate daily exercise; and healthy eating. The balance section is concerned with the frequency of intake of milk and dairy products, fruit, snacks, and water. The diversity section covers the intake of eggs, fish, vegetable side dishes, and soy and tofu, as well as the frequency of daily meals and eating alone. The moderation section pertains to the ability to restrict sweet or greasy food, sweetened drinks, and ramen.

For each domain, scores are classified as follows: “high”, 75–100%; “medium–high”, 50–74%; “medium–low”, 25–49%; and “low”, 0–24% [21]. In our study, the balance, diversity, moderation, and dietary behavior scores were medium–low (41.14%), medium–high (56.09%), high (81.55%), and medium–high (58.41%), respectively (Table 4). The overall NQ-E average was medium–high (60.88%).

Both food knowledge and the NQ-E score correlated significantly with food literacy (*p* < 0.05). Food knowledge and the NQ-E score increased by 0.026 and 0.245 points on average, respectively, for each 1-point increase in the food literacy score (Table 5).

## 4. Discussion

Research on health literacy is slowly ongoing in Korea, but there has been little focus on nutritional and food literacy [22]. However, outside of Korea, the use of the concept of “food literacy” has significantly increased in research [23]. Moreover, the results of various interventions have been implemented widely over the past decade. There is body of evidence that suggests improved food security and food literacy skills are possible outcomes of such interventions [24]. Since no tools were available in Korea to assess the food literacy for elderly to date, we could not identify the barriers and/or enablers for the elderly in utilizing their food literacy. Therefore, in this study, we developed and successfully validated 13 food literacy questionnaires focused on the elderly.

Tools for evaluating food literacy can be valuable. For example, interventions aimed at helping individuals prepare and cook food may reduce their risk of experiencing food vulnerability [24]. Previously, a food literacy questionnaire was developed for adults [16], and it systematically encompassed complex food literacy concepts by adding a food systems dimension (production, selection, preparation and cooking, intake, and disposal domains) to the existing literacy dimension (functional, interactive, and critical literacy domains). However, the fact that several different age-related changes could contribute to a decrease in overall literacy in older adults prompted us to develop a separate questionnaire specially developed for older adults. We considered the fact that the decreasing cognitive abilities in older people can contribute to a decrease in understanding, and, therefore, we tried to avoid questions about new topics and making a lengthy questionnaire [25].

Our questionnaire could be used as a simple tool for nutritionists to plan interventions aiming at increasing interest in various aspect of foods to reduce food vulnerability, or for the management of chronic diseases in the elderly, thus assisting them in leading healthy, autonomous lives.

The 13 food literacy questions developed in this study were spread into three domains: preparation and cooking, distribution, and production and disposal. Several questions were reclassified after factor analysis. For example, the question “I try to obtain accurate information about food and health” was initially classified as “intake” but was later reclassified to “preparation and cooking” after EFA. This suggests that, in practice, preparation and cooking can be considered as a single action, even though they comprise two apparently dissociable components. EFA also indicated that production and distribution are a single construct. Thus, there may be a disconnect between how the general public perceive such constructs and the theoretical classifications [16].

The NQ-E in this study was used as a comparison parameter to identify whether our developed food literacy scale somewhat matches the aspects of nutrition, such as dietary quality and food behavior. The food literacy scores on our questionnaire were highly correlated with the NQ-E scores in Korean elderly people, demonstrating the reliability of our food literacy measurement tool [16,21].

Understanding food literacy among the elderly for the development of customized diets and creating educational programs are essential to the prevention and management of chronic diseases in an aging population. According to the Korean National Health and Nutrition Examination Survey (KHHANES) adequate nutrition—including protein, fiber, water, vitamins, and minerals—may be more difficult to obtain for seniors who do not consume a balanced diet [26]. The NQ-E scores of our study, especially in the balance domain, were relatively low, suggesting that nutrition experts may tailor nutrition education with emphasis on improving the category of the NQ-E and/or food literacy scores.

A limitation of this study was the use of convenience rather than randomization sampling during the validation study, as well as the higher proportion of females versus male respondents. Particularly in research on the elderly, risk factors, educational background, income, living arrangements, and sex are important. Finally, the food knowledge questions we used to estimate the basal knowledge about food in general were not validated, which made it difficult to precisely determine the level of knowledge of the participants.

One might question the lack of questions that reflect Korean culture, but we focused more on the literacy of food production, cooking, and distribution, which might have different levels among countries and not much of cultural difference. Therefore, since most of the questions in our questionnaire were extracted from literature published globally, we suggest that our tool may be applicable not only to the Korean population, but also to populations worldwide. Additionally, another strength of this study is that we integrated a food system dimension into existing literacy measurement tools, putting knowledge and areas of action into a broader context.

## 5. Conclusions

In conclusion, we developed and validated a new “food literacy measurement tool” specialized for people over 65 years old. The validity and reliability of this questionnaire were proved by both expert and general populations, and a final questionnaire with a total of 13 items was produced. Tailored approaches to diet guidelines and educational programs based on the level of food literacy will be valuable for efficiently preventing and managing diseases in increasingly aging societies. This study provides a foundation to prevent and manage chronic diseases in the elderly and to help them lead autonomous, healthy lives.

## Figures and Tables

**Figure 1 ijerph-18-04979-f001:**
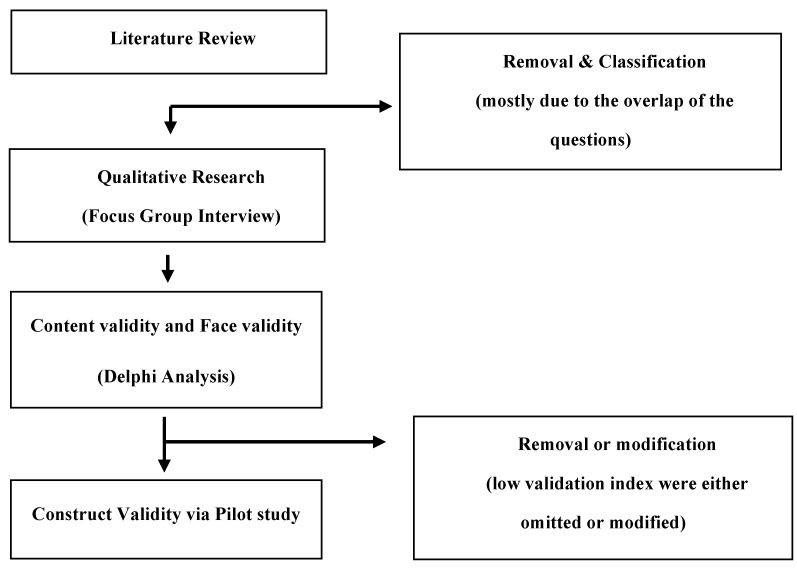
Flow chart of the development and validation of the food literacy questionnaire

**Table 1 ijerph-18-04979-t001:** Content validity ratio values of the final 32 questions.

Item	CVR ^¶^
I usually check a food’s country of origin	1.00
I usually check for GMO label on food products	0.87
I understand the agricultural food certification (organic, pesticide free, etc.) label	0.87
I can find information about food production, such as the “animal welfare” certification for meat and eggs	0.73
I am aware of how the method of producing agricultural and livestock impacts the environment	0.60
I usually check the shelf-life of foods	1.00
I can find information on methods of food distribution including local food	0.87
I am aware of how different food transportation methods impact the environment	0.73
I can buy food efficiently	0.20
I can buy as much food as I need	0.73
I choose food considering how easily I can swallow and digest it	1.00
I try to choose new foods, not only the ones that I am used to eating	0.73
I can find foods that suit my health and circumstances (based on costs, etc.)	0.87
I can look up or enquire about the various ways to determine the quality (taste, freshness, etc.) of food	0.87
I can talk about the pros and cons of Korean foods	0.87
I can determine whether food is necessary for me by watching/reading food advertisements	0.87
I usually cook and store food with care to avoid food poisoning	1.00
I can prepare nutritionally balanced meals	1.00
I store food in a way that maintains its quality	1.00
I often discuss or ask about the various ways to cook	1.00
I can determine the condition of foods by monitoring the preparation and cooking process	1.00
I try to obtain accurate information about food and health	1.00
I usually try to eat a variety of food groups, including grains, fish, meat, vegetables, fruits, milk, etc.	1.00
I ascertain the importance of foods by referring to dietary guidelines	0.87
I change my diet for prevention and management of diet-related diseases such as obesity, high blood pressure, and diabetes	0.87
If I have any questions about foods, I can obtain accurate information from experts or reliable organizations	0.87
I take health or nutritional supplements after discussing them with doctors or experts	0.87
I can evaluate my usual diet	1.00
I can get food information through the TV, newspapers, etc. and determine food suitability according to my situation	1.00
I am usually aware of reducing food waste	1.00
I am aware of the appropriate way of recycling food packaging	0.87
I am aware of the environmental impact of food waste and know how to dispose it	1.00
Average	0.88

^¶^ Content Validity Ratio.

**Table 2 ijerph-18-04979-t002:** General characteristics of the participants (*n* = 205).

Categories	Total (%)
**Age (years)**	
65–74	141 (68.8)
75–84	53 (25.9)
≥85	11 (5.4)
**Sex**	
Male	83 (40.5)
Female	122 (59.5)
**Education**	
≤middle school	114 (55.6)
High school	57 (27.8)
College	26 (12.7)
≥Graduate school	6 (2.9)
**Monthly income** (KRW ^1^**)**	
<2,000,000	87 (42.4)
2~4,000,000	80 (39.0)
4~6,000,000	21 (10.2)
≥6,000,000	15 (7.3)
**Average number of household**	
1	26 (13.7)
2	113 (55.1)
3~4	49 (23.9)
5~6	17 (8.4)

*n* (%) Numbers in brackets present percentage of participants. ^1^ Korean won.

**Table 3 ijerph-18-04979-t003:** Results of the EFA ^¶^ on the final questionnaire (*n* = 205).

Factor	Item		Communalities	Factor Loading		
				1	2	3
Preparationandcooking	I can discuss and enquire about various ways to cook.		0.684	0.794	0.232	0.021
	I can prepare nutritionally balanced meals.		0.726	0.786	0.294	0.145
	I store food in a way that maintains food quality.		0.761	0.764	0.145	0.395
	I can judge whether food is hygienic based on the preparation and cooking processes.		0.648	0.759	0.197	0.181
	I usually cook and store food with care, as I am cautious about food poisoning.		0.734	0.721	0.089	0.453
	I try to obtain accurate information about food and health.		0.565	0.630	0.389	0.128
Distribution	I am aware of how methods of producing agricultural and livestock impact the environment.		0.744	0.197	0.834	0.100
	I am aware of how different food transportation methods impact the environment.		0.692	0.258	0.766	0.198
	I can find information on food distribution methods, including for local food.		0.631	0.233	0.752	0.108
	I can find information about food production, such as the “animal welfare” certification for meat and eggs.		0.529	0.161	0.698	0.122
Productionanddisposal	I usually check a food’s country of origin.		0.681	0.140	0.180	0.793
	I usually check the agricultural certification of foods (organic, non-agricultural, etc.).		0.714	0.136	0.421	0.720
	I usually try to reduce food waste.		0.554	0.278	0.006	0.690
Eigenvalue			-	27.892	50.228	66.622
Cumulative % of variance			-	5.910	1.542	1.208

^¶^ Explanatory Factor Analysis.

**Table 4 ijerph-18-04979-t004:** NQ-E **^¶^** analysis results.

Domain	Range	Total (*n* = 205)
Balance	0–100	45.14
Diversity	0–100	56.09
Moderation	0–100	81.55
Dietary behavior	0–100	58.41
Total NQ-E score	0–100	60.88

^¶^ Nutrition Quotient for Elderly.

**Table 5 ijerph-18-04979-t005:** Linear regression analysis.

Variables	Coef.	SE	*p*-Value	95% CI	
Food knowledge score andfood literacy score	0.026	0.004	<0.001	0.019	0.033
NQ-E score andfood literacy score	0.245	0.026	<0.001	0.194	0.295

Coef. = coefficient; CI = confidential interval; SE = standard error.

## Data Availability

The study did not report any data.

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
