# Peer review of "Development and Validation of a Food Literacy Assessment Tool for Community-Dwelling Elderly People"

_ijerph, 2021, doi:10.3390/ijerph18094979_

Round 1

Reviewer 1 Report

Abstract: line 25 insert "Korean" elderly.

Figure 1. By What criteria were items removed/added?

Will the questionnaire be applicable to other cultural groups? (beyond Korean)

Line 66: questionnaire should be plural

2.2.1 Figure did not print clearly

Lines 106-107: revise English

Table 1. Items seem excellent. Was there an adjustment for multiple comparisons? Why were so many removed from the final questionnaire?

Line 134: English

Lines 193-196: English

Lines 223-225: How did they evaluate? Diet recall? Food diary? Other? Explain.

Lines 234-238: Discuss cultural implications of using items derived from other cultures applied to Korean elderly. Discuss need for additional studies among Korean elderly to further explore cultural validity. 

Author Response

Response to Reviewer 1 Comments

Point 1: Abstract: line 25 insert "Korean" elderly.

Response 1: We inserted that word.

Line 24 : elderly -> Korean elderly

Point 2 : Figure 1. By What criteria were items removed/added? 

Response : We added sentence ‘mostly due to the of overlap of the questions’ in figure 1. 

Line 75-55 : Added ‘mostly due to the of overlap of the questions ‘

Point 3 : Will the questionnaire be applicable to other cultural groups? (beyond Korean)’

Response : The questionnaire was extracted from the literature developed overseas. The final 13 questions are applicable to any cultural background. And honestly, we believe that is the strength of our questionnaire.

Point 4: Line 66: questionnaire should be plural

Response : We corrected as suggested

Line 65 : questionnaire -> questionnaries

Point 5: 2.2.1 Figure did not print clearly

Response : We deleted the inserted image and changed to MS word format.

Point 6:  Lines 106-107: revise English

Response : We deleted which were repeated. -> ‘Cronbach’ alpha and exploratory factor analysis and Kaiser- Meyer-Olkin’

Point  7:  Table 1. Items seem excellent. Was there an adjustment for multiple comparisons? Why were so many removed from the final questionnaire?

Response :  Unfortunately we did not perform multiple comparison analysis, which is to my knowledge not used very frequently. Do you think it will help for the explanation?

Items with CVR less than 0.49 in Delphi test, and items that did not meet the criteria (communalities extraction < 0.4, factor loading < 0.4) in confirmatory factor analysis were deleted and ended up with 13 questions.

Point 8:Line 134: English

Response (original version): The general characteristics of the participants are shown in Table 2. More female than man (59.5% vs. 40.5%) answered the survey with the mean age of 71.80 ± 6.81 years.

│ (Changed version) The general characteristics of the participants are shown in Table 2. The number of female participants who answered the survey (mean age of 71.80 ± 6.81 years), was greater than that of male participants (59.5% vs. 40.5%).

Point 9: Lines 193-196: English

Response (original version):  Outside Korea, however, the concept of “Food literacy” has significantly increased in research and result of various intervention became widely implemented over the past decade [22]. There is body of evidence to suggest that improved food security and food literacy skills can result from those interventions.

│ (Changed version) However, outside of Korea, the use of the concept of "Food literacy" has significantly got increased in research. Also the result of various intervention was implemented widely over the past decade. There is body of evidence that suggests improved food security and food literacy skills is possible outcome through those interventions.

Point 10: Lines 223-225: How did they evaluate? Diet recall? Food diary? Other? Explain.

Response : I inserted the reference which is a national survey data. 

(Original version):  Adequate nutrition—including protein, fiber, water, vitamins, and minerals—may be more difficult to obtain for seniors who do not consume a balanced diet.

│ (Changed version) According to the Korean National Health and Nutrition Examination Survey (KHHANES), adequate nutrition—including protein, fiber, water, vitamins, and minerals—may be more difficult to obtain for seniors who do not consume a balanced diet.

Point 11: Lines 234-238: Discuss cultural implications of using items derived from other cultures applied to Korean elderly. Discuss need for additional studies among Korean elderly to further explore cultural validity. 

Response :  We agree that there must be some items included that reflects Korean culture. However, when we performed the Delphi survey, no specific items were suggested by the expert, suggesting that the questions were quite suitable for all countries. Will include some suggestions in the discussion.

(Original version):  Despite these limitations, since most of the questions in our questionnaire were extracted from literature conducted outside Korea, we suggest that our tool may be applicable not only to Korean, but also to populations worldwide. 

│ (Changed version)  One might question about not including any question that reflects Korean culture. But we focused more on the literacy of food production, cooking and distribution which might have a different level among countries, but, not much of cultural difference.   Therefore, since most of the questions in our questionnaire were extracted from literature published globally, we suggest that our tool may be applicable not only to Korean, but also to populations worldwide. 

Reviewer 2 Report

The manuscript entitled ‘Development and Validation of a Food Literacy Assessment Tool for Community-dwelling Elderly people’ is a trial to develop a food literacy questionnaire that can be used to improve healthcare of elderly people. Based on the set of articles from the year 1960 till  2020, after several stages of the selection process, 13 questions were finally identified. The problem to select the most significant set of questions can be an important tool to support food production and the paper address that issue. Unfortunatelly, there are several aspects that should be improved before publication in ‘International Journal of Environmental Research and Public Health’.

COMMENTS:

1. Please provide the number of the papers found (l. 65-66).

2. Please provide the description to justify the experience, skills and the reason of chosen number of experts used in the stage of extraction of preliminary questions. Please describe the rules of the extraction process.

3. The ‘Conclusions’ section should be improved by the main findings from the substantive analysis of the set of the finally selected 13 questions.

4. The ‘Discussion’ section should be improved by the extended overview through the results published in the available literature.

5. The provided analysis is based only on the selected range (more than 65 years) and there is no comparison with other questionnaires. The additional analysis for the scope between 15 and 65 years should be added as well as for the questionnaires developed in literature previously.

6. To justify the statement of the significance of obtained results (l. 235-236 – ‘our tool may be applicable not only to Korean, but also to populations worldwide’), the comparison between the set of questions identified in The Self-Perceived Food Literacy Questionnaire and the set of questions identified in the provided research should be delivered.

7. Please reformulate ‘Conclusions’ section to present the main findings of the research in a concise and precise way

Author Response

Thank you for your comments.

Round 2

Reviewer 2 Report

Dear Authors,

The manuscript is suitable for publication in its current form.